# Development of a New Method for the Quantitative Generation of an Artificial Joint Specimen with Specific Geometric Properties

**Seungbeom Choi, Sudeuk Lee, Hoyoung Jeong and Seokwon Jeon *** 

Department of Energy Systems Engineering, Seoul National University, Gwanak-gu, Seoul, Korea; chbum092@snu.ac.kr (S.C.); ics1961@snu.ac.kr (S.L.); hyjung04@snu.ac.kr (H.J.)

*   Correspondence: sjeon@snu.ac.kr; Tel.: +82-2-880-8807

**Abstract:** A rock joint is a planar discontinuity that has significant influence on the mechanical and hydraulic characteristics of rock mass. Laboratory experiments are often conducted on a joint to investigate and provide fundamental information for rock mass analysis. Although joint roughness and mechanical aperture exert great effects on the experimental results, controlling them in quantitative manner is quite complicated and consumptive in terms of specimen preparation. A new and simple method for the quantitative generation of the joint specimen was proposed in this study. Based on random midpoint displacement method, a joint specimen with a void space inside was generated. Parametric studies for the roughness and mechanical aperture were carried out, and as a result, the two joint properties could be controlled by manipulating input parameters of random midpoint displacement method. In order to validate the proposed method, two joint specimens, which had different levels of roughness and aperture, were generated and printed. Surface coordinates of the specimens were obtained by a 3D laser scanner, and calculated to make a comparison between the target values and the estimated values. Results showed that the method was capable of generating joint specimens with satisfactory precision.

**Keywords:** artificial joint specimen; roughness; mechanical aperture; quantitative generation; random midpoint displacement method; 3D printer

## 1. Introduction

A rock joint is a planar discontinuity that has little tensile strength and it exerts influence on mechanical behavior of rock mass, since it acts as a weak plane. At the same time, a joint has much higher hydraulic conductivity than the rock matrix, so that the majority of fluid flow in the rock mass happens through the joint [1–3]. Therefore, it is of great importance to accurately understand the characteristics of a joint in many engineering applications, as it affects both the mechanical and hydraulic behaviors of the rock mass. For examples, in engineering projects, such as rock slope design, underground storage facilities, the repository for radioactive disposal [4,5], and geothermal energy production [5,6], etc., it is necessary to consider the effect of the joint on the stability and efficiency of the projects. Furthermore, new technologies for coping with global climate change have been developed recently, including geological storage of $CO_2$. There is some possibility that $CO_2$ may leak out and flow through the joint, which should be taken into account for guaranteeing the feasibility and stability of the storage.

In situ measurements can derive the hydraulic characteristics of rock mass, which consider the effects of multiple joints or joint sets. However, they are sometimes consumptive in terms of time and expenses, so that the features are investigated by laboratory experiments on a joint, providing

fundamental properties for rock mass analyses. A joint has several relevant properties, such as roughness, aperture, stiffness, contact area, matedness, and so on [7,8]. They exert complex influences on the mechanical and hydraulic characteristics of the joint, interacting with each other. Mechanical aperture is a separation between the two corresponding joint walls. It is defined as a distance between those walls in the direction perpendicular to the reference plane at a point [7]. The aperture acts as flow path in a joint, thus, several researches have investigated the relationship between aperture and other properties [9–11]. Roughness has drawn attention for the relationship, since it is a widely-used engineering parameter, and relatively easy to be measured. The roughness is usually defined by its height distribution or shape, and there are many quantitative roughness parameters. For instance, Tse and Cruden (1979) proposed (center line average), (mean square value), (root mean square), and $Z_2$ (the *RMS* of the first deviation of the profile) [12]. Barton and Choubey (1977) proposed the *JRC* (joint roughness coefficient) with 10 standard joint profiles, based on numerous shear test results [13,14]. Even though *JRC* is not a quantitative parameter, it has been widely used in engineering projects, mainly due to its simplicity and practicability. The effects of aperture and roughness are of importance in the mechanical and hydraulic behaviors of a joint, so that an accurate understanding of them is quite necessary.

In many cases, artificial joint specimens are prepared, in order to carry out mechanical and/or hydraulic experiments. The joint specimens are often created mechanically by using a splitter or a V block, which applies a concentrated load at the center of rock block or cylinder similar with Brazilian jaws [15,16]. Tensile stress generated along the loading axis bisects the rock into two halves. Although it is a simple method to create a joint specimen, it requires excessive efforts if a specific joint roughness is desired. Therefore some experimental works have derived results from limited test conditions in terms of the joint roughness [3,16]. The reproducibility of this approach is hardly guaranteed so that casting method, which uses cement mortar or epoxy, is adopted as an alternative to make multiple joint specimens that have similar levels of roughness [17,18]. The casting method mitigates the reproducibility problem to some extent; however, it also has a shortcoming. The aperture, which is an important joint property along with the roughness, cannot be controlled by the method. Therefore, the geometric conditions of a joint, i.e., the roughness and aperture, could not be controlled quantitatively, and simultaneously in many cases.

In this study, a new and simple method was proposed to generate an artificial joint specimen. This method enables to control the roughness and aperture at the same time in a quantitative manner. Two corresponding joint surfaces were generated by fractal theory, specifically by the random midpoint displacement method, producing point-cloud type data of the surfaces. By using proper input parameters, the roughness as well as the aperture of a joint could be controlled and manipulated. The generated data can be directly utilized in the theoretical approach or numerical simulation, since they provide the x, y, and z coordinates of the joint surfaces. Thanks to recent developments in 3D printing technology, this method is even capable of being adopted in experimental works, overcoming the limitations that have been explained above.

## 2. Background Theory

Several methodologies have been proposed to make an artificial joint surface. The Fourier transformation can be used to make and evaluate the rough surfaces [19] by manipulating the number of harmonics to be calculated, and the Fourier coefficients. Stochastic functions such as the autocorrelation function can also be applied to this procedure [20] by controlling the range and standard deviation of the joint height. The fractal Brownian function is one of the viable options to make synthetic surfaces [21] with many applications being reported. Xie (1993) [21] stated that the rock joint could be successfully simulated by fractal theory in terms of roughness, aperture, pore, permeability, and so on. Natural rock joints usually show self-affinity, which is a feature that different amount of scaling should be considered in transformation. The fractal Brownian functions could take the feature into account; thus it can be applied in estimating and/or generating the artificial joint surfaces.

Among them, the random midpoint displacement method was adopted in this study to make point-cloud type data of the joint specimens. It has several benefits of application. First, it is quite straightforward, and its outcomes can be directly imported to a 3D printer. Also, the manipulation of input parameters can derive satisfactory results in controlling the joint roughness and aperture. The algorithm of the random midpoint displacement method is explained briefly in the following procedure (Figure 1).

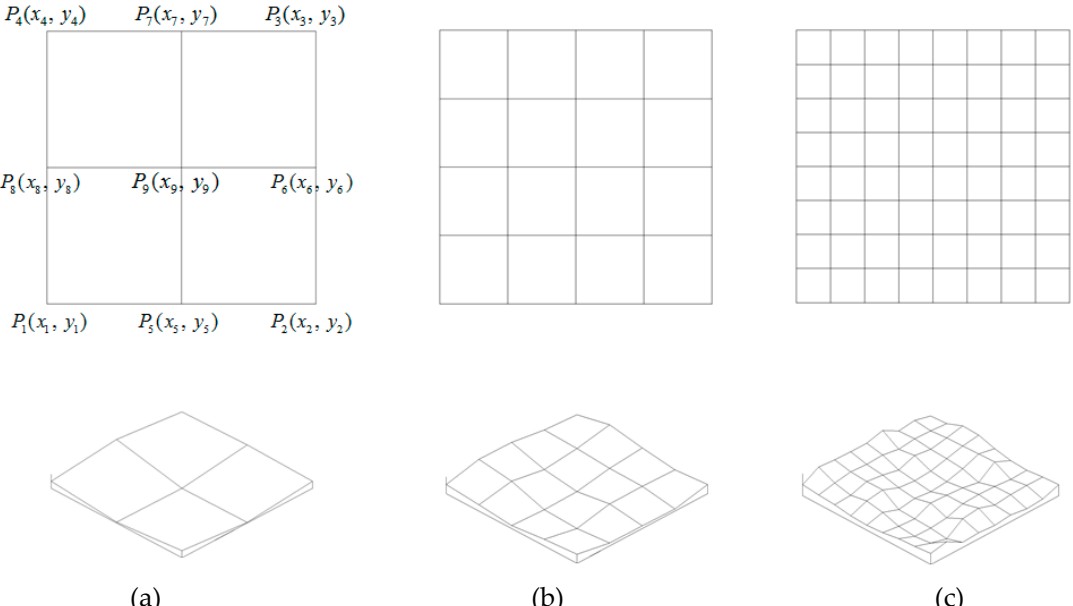

**Figure 1.** The generation sequence of a rough joint surface following the random midpoint displacement method: (a) Joint shape when $GL = 1$, with nine points on the joint surface; (b) Joint shape when $GL = 2$; (c) Joint shape when $GL = 3$ (After Seo and Um [22]).

For the sake of brevity, let us assume a square joint surface that has a certain side length ($L$). Then, the elevation of the four corner points should be determined first. If the points have the same elevation, it is called the stationary profile, and if not, it is called non-stationary. The interval between the neighboring points is calculated as Equation (1).

$$\Delta x = \Delta y = \frac{L}{2^{GL}} \tag{1}$$

where $GL$ (generation level) is the number of generation sequences, which also defines the resolution of a joint surface.

Based on the four predetermined corner points, coordinates of midpoint ($P_9(x_9, y_9)$) in Figure 1a are calculated by Equation (2).

$$\begin{aligned} x_9 &= \tfrac{1}{4}(x_1 + x_2 + x_3 + x_4) \\ y_9 &= \tfrac{1}{4}(y_1 + y_2 + y_3 + y_4) \\ z_9 &= \tfrac{1}{4}(z_1 + z_2 + z_3 + z_4) + D_1 \end{aligned} \tag{2}$$

where $D_1$ is the Gaussian random number, which follows a normal distribution that is defined by a couple of input parameters, and $GL$ (as presented in Equation (3)).

$$N\left(0, \frac{\sigma^2(1 - 2^{2H-2})}{(2^{GL})^{2H}}\right) \tag{3}$$

where $\sigma$ is a term that is related to the amplitude of point elevation, and $H$ is the Hurst exponent. In a three-dimensional case, $H$ has a relationship with fractal dimension ($D$), which is $D = 3 - H$. If the midpoint is located on a side line of the square, only two neighboring points are included in the calculation. By iterating this procedure with desired $GL$ value, point-cloud type data of an artificial surface can be easily generated.

## 3. Development of a New Method and its Parametric Study

### 3.1. Development of a New Method for Generating Three-Dimensional Joint Specimens

As explained above, a three-dimensional joint surface could be easily generated by adopting the random midpoint displacement method. Based on this, a whole three-dimensional joint specimen, which includes two corresponding joint surfaces and a void space (collection of apertures) inside, could be generated as well. Figure 2 shows a conceptual procedure for generating a joint specimen, but it is drawn in two dimensions for clear understanding.

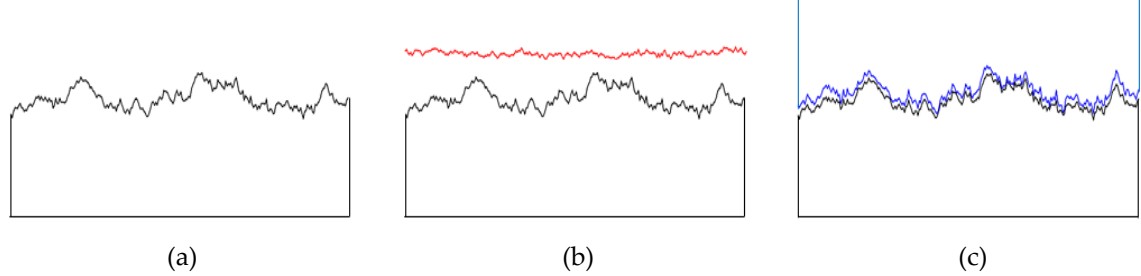

|(a)|(b)|(c)|

**Figure 2.** Conceptual procedure for generating a joint specimen that includes a void space inside.

First, a lower joint surface is generated according to the random midpoint displacement method (Figure 2a). The roughness of the lower surface can be controlled quantitatively by manipulating the input parameters. Then, a void space is generated separately by the same method (Figure 2b). The generated data can be treated as a joint surface, but at the same time, it can represent the upper bound of a void space. Thus, after generating point-cloud data of the void space then they are translated in parallel, so as to make the minimum elevation (z coordinate) of the void space zero. Each elevation of the points means the aperture of that point, which also can be controlled in a quantitative manner. At last, the lower surface and void space is summed to make an upper joint surface (Figure 2c). Figure 3 shows a detailed flowchart.

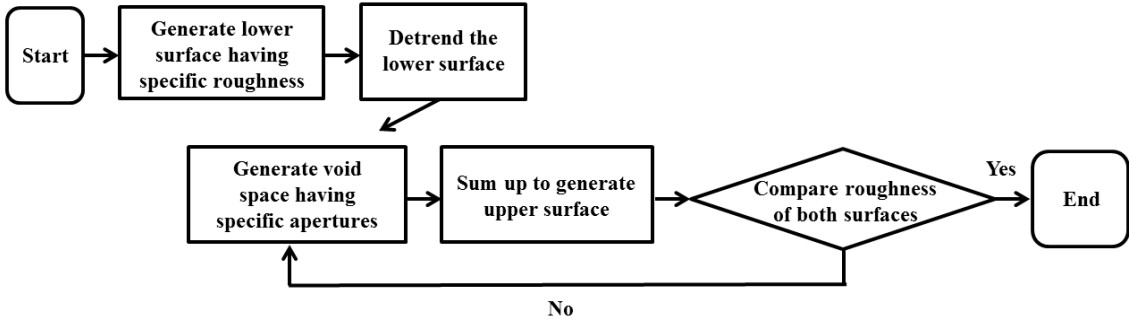

**Figure 3.** Flowchart for generating a joint specimen in a quantitative manner.

The generated surface may show a inclination at some degree. When it comes to evaluating the roughness quantitatively, this inclination trend should be removed. Therefore, multilinear regression was included in the flowchart so as to detrend the lower surface [3,23]. Since all coordinates of the lower

surface ($x_0$, $y_0$, $z_0$) are known, an optimum regression plane could be found, based on the coordinates. Subtracting a trend plane ($z_{trend}$) from the original coordinates gives the final lower surface:

$$z = z_0 - z_{trend} \ where \ z_{trend} = n_1 x_0 + n_2 y_0 + n_3 \tag{4}$$

Meanwhile, two corresponding joint surfaces usually show a similar level of roughness, unless they are highly weathered. Thus, after generating the upper surface, a comparison of the roughness between upper and lower surfaces was included in the flowchart. If they show similar roughnesses, the procedure would end, and if not, the void space (aperture distribution) would be generated again to make another point-cloud, but with the same distribution characteristics.

### 3.2. Parametric Study for Generating Joint Specimen

It is known that the roughness of a joint surface generated by the random midpoint displacement method is affected by joint length ($L$), generation level ($GL$), asperity amplitude ($\sigma$), and the Hurst exponent ($H$) [22,24]. However, considering the testing environment, such as the size of a specimen and the resolution of a 3D printer, it is reasonable to assume that $L$ and $GL$ are actually determined before the surface generation. Therefore, the roughness is determined according to a combination of $\sigma$ and $H$. The Hurst exponent, which is related to the fractal dimension, has a range of 0~1, and although there is no limit on the amplitude, it was set to 0~10 in this study.

The joint roughness can be classified into waviness and unevenness. The former defines relatively large, macroscopic scale and the latter means relatively small, microscopic scale roughness. In order to investigate the effect of input parameters, i.e., $\sigma$ and $H$, on the roughness, FFT (Fast Fourier Transform) analyses on some joint profiles were carried out. Several joint profiles were generated with a fixed joint length of 100 mm and a generation level of 7. The amplitude ($\sigma$) varied from 1 to 10 with an interval of 1, and the Hurst exponent varied from 0 to 0.9 with the interval of 0.1. Among those profiles, six examples are presented in Figure 4.

Figure 4a–c (in the first column) shows the joint profiles and their FFT results under the conditions of varying Hurst exponents and a fixed amplitude, while Figure 4d–f (in the second column) show the results under varying amplitudes and a fixed Hurst exponent. The red column in Figure 4 means the normalized amplitude, so that its sum is 100, and the blue curve means the cumulative value. Judging from the profiles in Figure 4a–c, the joint roughness decreased with the increase of $H$. Also, at the same time, the graphs in Figure 4a–c show that portion of low frequency, i.e., the microscopic asperity, increased remarkably when the $H$ increased. On the other hand, the profiles in Figure 4d–f showed that the roughness increased with the increase of $\sigma$, while the FFT results showed little difference. However, it is noteworthy that profiles in Figure 4d–f had larger scales in the y-axis than those in Figure 4a–c. Therefore, it could be concluded that the increased value of $H$ made the overall roughness of a joint smoother, but it also made the portion of microscopic asperity larger, while $\sigma$ made the overall roughness and the absolute height of a joint larger.

Several quantitative roughness parameters have been proposed; however, most of them are two-dimensional parameters. It is obvious that the representativeness of a joint profile is not enough for a whole joint surface, since the joint itself is a three-dimensional structure. In order to represent the characteristics of a joint better, a few three-dimensional roughness parameters have been proposed [25,26]. They can guarantee more representativeness, and they take joint anisotropy into consideration; however, their applicability and practicality could not have exceeded those of the two-dimensional parameters.

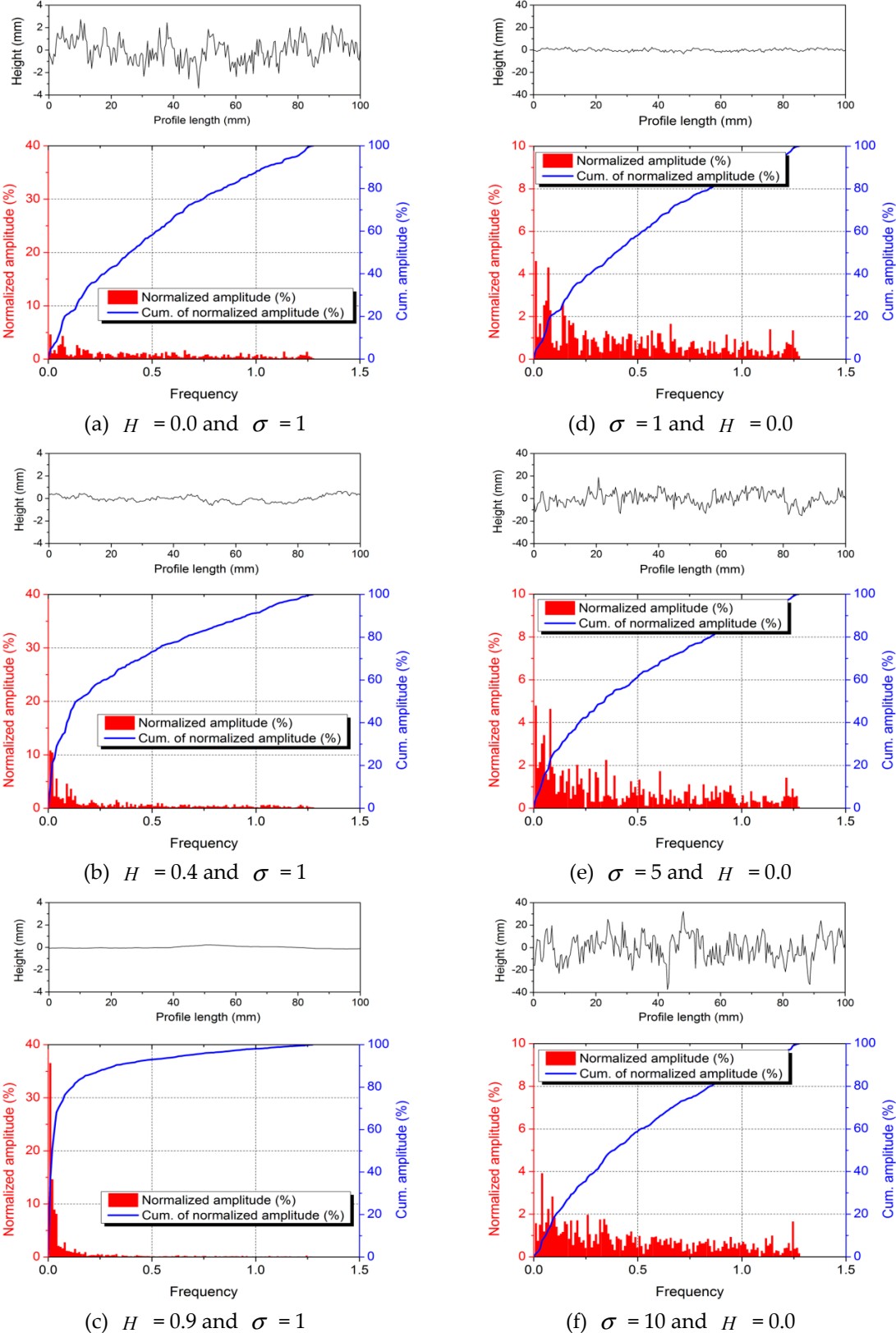

**Figure 4.** Joint profiles generated by a combination of input parameters ($\sigma$, $H$) and their Fast Fourier Transform (FFT) results. The red column denotes a normalized histogram, and the blue curve denotes the cumulative curve of the histogram.

In this study, a compromise method was adopted to determine the joint roughness. Figure 5 shows a schematic diagram for this determination. Let us assume that a long axis (the x direction in Figure 5) of the rectangle corresponds to shear direction. Then, profiles defined in the xz plane are of interest. Additionally, the moving xz plane along the y axis would define several joint profiles (Figure 5b). Each and every profile within the xz plane is obtained and calculated by following one of the quantitative two-dimensional parameters. Then, a distribution of the two-dimensional parameter could be calculated, and the average of the distribution was defined as a typical value. Tse and Cruden (1979) [12] suggested $Z_2$ (Equation (5)) and it was selected as an estimator in this study.

$$Z_2 = \sqrt{\frac{1}{L}\int_0^L \left(\frac{dy}{dx}\right)^2 dx} = \sqrt{\frac{1}{L}\sum_{i=1}^{n-1}\frac{(y_{i+1}-y_i)^2}{x_{i+1}-x_i}} \tag{5}$$

Several joint surfaces were generated according to the combinations of input parameters. The length of joint ($L$) and the generation level ($GL$) were set to 100 mm and 7, respectively. The amplitude ($\sigma$) varied from 1 to 10 with the interval of 1, and the Hurst exponent varied from 0 to 0.9, with an interval of 0.1. Then, another set of point-cloud data, which represented the void spaces, was generated under the same input parameters. Since the aperture values were calculated at each and every point of space, a distribution of apertures was obtained as well. The average was selected as a typical value, and the minimum elevation was set to 0. A three-dimensional plot of roughness ($Z_2$) and mechanical aperture are shown in Figure 6.

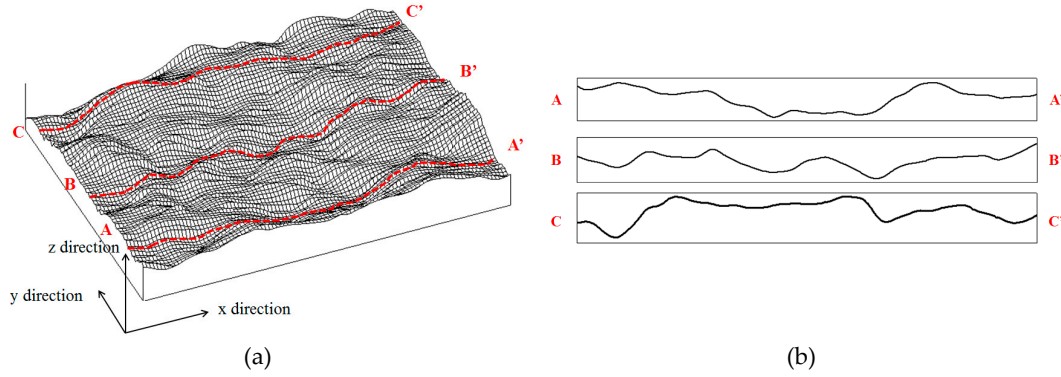

(a)　　　　　　　　　　　　　　　　　　　　　　　　　　　　　　　(b)

**Figure 5.** Schematic diagram for determining the joint roughness adopted in this study: (**a**) Three-dimensional plot of a joint; (**b**) Obtained profiles along a certain direction.

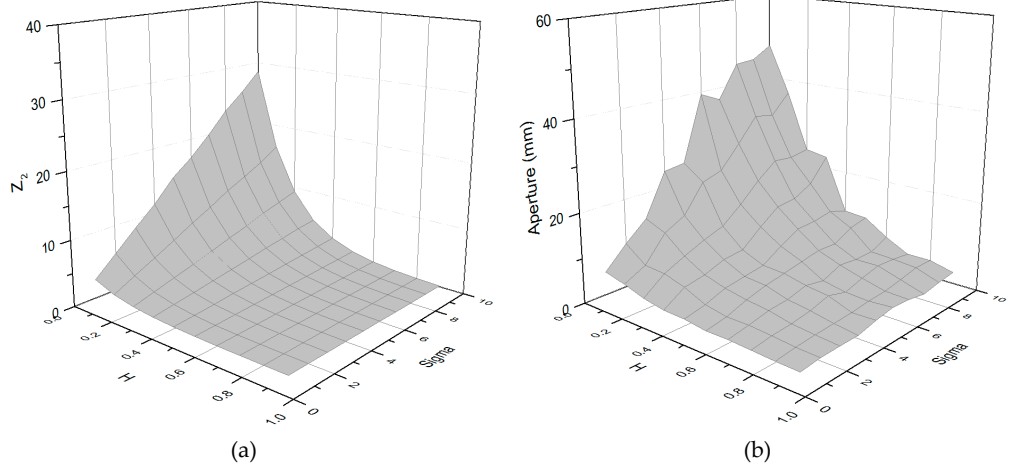

(a)　　　　　　　　　　　　　　　　　　　　　　　　　　　　　　　(b)

**Figure 6.** Three-dimensional plots from the parametric study; (**a**) results of the roughness variation ($Z_2$); (**b**) results of the mechanical aperture variation.

As it can be seen in Figure 6a, the roughness of a joint increased with the increase of $\sigma$ and the decrease of $H$. This was the same tendency that was presented in Figure 4. Therefore, under the predetermined values of $L$ and $GL$, the combinations of the two input parameters could be constructed, and eventually a desired joint roughness could be obtained by properly selecting a combination. Figure 6b shows a variation of mechanical aperture, according to the input parameters. The mechanical aperture increased with the increased of $\sigma$ and the decrease of $H$. It is noteworthy that the magnitude of the mechanical aperture in Figure 6b was quite large in some cases. In those cases, an appropriate scaling should be considered, so as to obtain the desired void space.

## 4. Verification and Discussion

In order to validate the proposed joint-generating method, some verification works were carried out. Two joint specimens, which had specific target geometric characteristics, were generated by the proposed method, and then printed by a 3D printer. The printed specimens were scanned by 3D laser scanner, so as to obtain surface information, which was further used to measure the roughness and aperture. Then, verification was conducted by comparing the estimated results and the original target values. Table 1 shows the target geometric properties of the joints.

**Table 1.** Target geometric properties of the joint specimens.

|  | Roughness (*JRC*) | Initial Mechanical Aperture (mm) |
|---|---|---|
| Case 1 | 4~6 | 0.1 |
| Case 2 | 14~16 | 0.5 |

As mentioned earlier, the applicability of *JRC* exceeded those of the other roughness parameters, even though it was rather qualitative. Thus, the target roughness values were expressed in *JRC*. Tse and Cruden (1979) [12] suggested a relationship that could convert $Z_2$ into *JRC* (Equation (6)):

$$JRC = 32.2 + 32.47 \log Z_2 \tag{6}$$

Contrary to the $Z_2$, *JRC* has a limited range, from 0 to 20. Therefore, the domain $Z_2$, should be bounded as well, when using Equation (6), which is roughly from 0.10 to 0.42. After obtaining three-dimensional plots in Figure 6 under the predetermined values of the lengths and generation level, combinations of $\sigma$ and $H$ were selected for each case. The length and generation level were set to 100 mm and 7, respectively. The lower surface of case 1 was generated under the conditions of $\sigma = 6$ and $H = 0.8$, and the void space was generated under the same input parameters, but scaled down to achieve the target aperture value. At the same time, the lower surface and the void space of case 2 were generated by the same procedure under the conditions of $\sigma = 7$ and $H = 0.7$. Figure 7 shows the printed joint specimens and their 3D models, which were generated under the same coordinates.

Then, the surfaces of the two joint specimens were scanned by a 3D laser scanner. The scanned joint area was 100 mm × 100 mm, and the scanning interval was 0.5 mm. The roughness was evaluated by the procedure explained above, and the aperture was measured by the superposition method [3,23]. The superposition method was capable of utilizing the scanned three-dimensional surface data, which were already obtained when determining the joint roughness. A brief explanation of the superposition method is as follows. First, the point data of both the lower and upper surfaces are arranged vertically by parallel and symmetric translation. Then, the upper surface is moved in parallel downward step-by-step, to make sure that it meets a certain criterion of aperture estimation. When the criterion is met, the distance of all of the corresponding points is calculated to estimate its distribution.

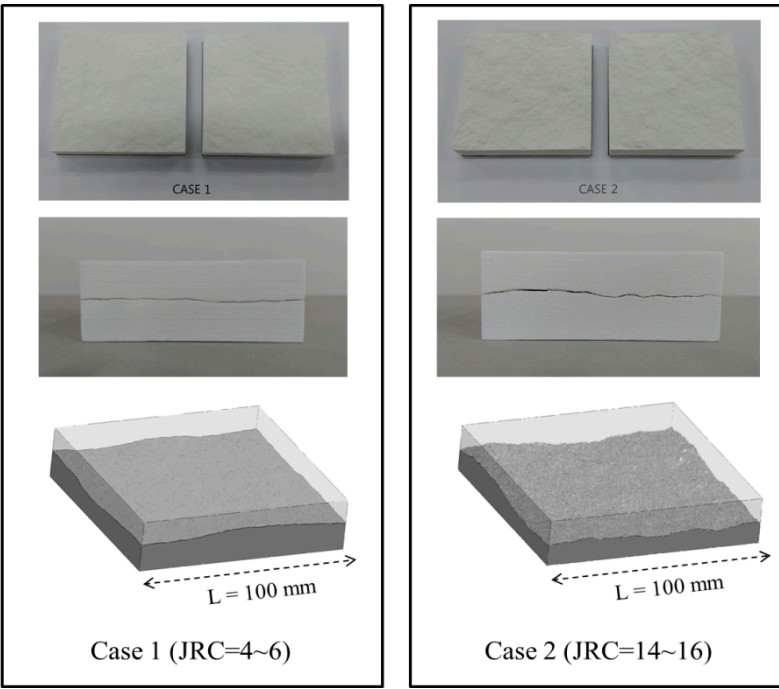

**Figure 7.** Picture of the printed specimens and their 3D models (side length = 100 mm).

The criterion for determining the mechanical aperture was defined when 1% of the total surface points were in superposition. It was referenced from experimental results that the area contacted by the dead weight of joint was approximately 1% of the total surface [27]. Since only gravitational force was applied to the specimen, it is called the initial mechanical aperture. Although the number of points did not directly determine the area, it was assumed that the ratio of points was equal to the ratio of the area, since a sufficient number of points was generated. When superposing, the upper surface was translated downward by 0.001 mm at a step until 1% superposition was reached. Then, the distribution of the aperture was calculated, and every aperture of the superposed points was set to zero. The calculated distributions of *JRC* and the initial mechanical aperture are shown in Figure 8, and the statistics of the distributions are presented in Table 2. The curves in Figure 8 mean a normal distribution curve that is fitted based on the estimated results.

As can be seen in Figure 8a,b, the roughness (*JRC*) roughly followed a normal distribution. Judging from the average of each case, the original target roughness was achieved quite well by the proposed method. At the same time, the estimated initial mechanical aperture also showed a good consistency with the target value. However, little discrepancy was found between them, and this was because the initial mechanical aperture was calculated when 1% of surface points were superposed. It could be mitigated with a couple of trial and error approaches, if some margin for the superposition would be included when scaling down the data of the void space. Distributions of aperture (Figure 8c,d) could be presented as the normal distribution. Natural rock joints usually followed a normal distribution or log-normal distribution [4,23,28] as well, and the same tendency could be found in this study. This result could be explained by the generation algorithm, where the Gaussian random number ($D_1$, in Equation (2)) was involved. Since the z coordinate was calculated following a Gaussian distribution, the roughness and aperture showed some normality in nature. Therefore, it could be concluded that the developed method was able to generate the joint surface in a quantitative manner, which could simulate similar characteristics of the natural rock joint.

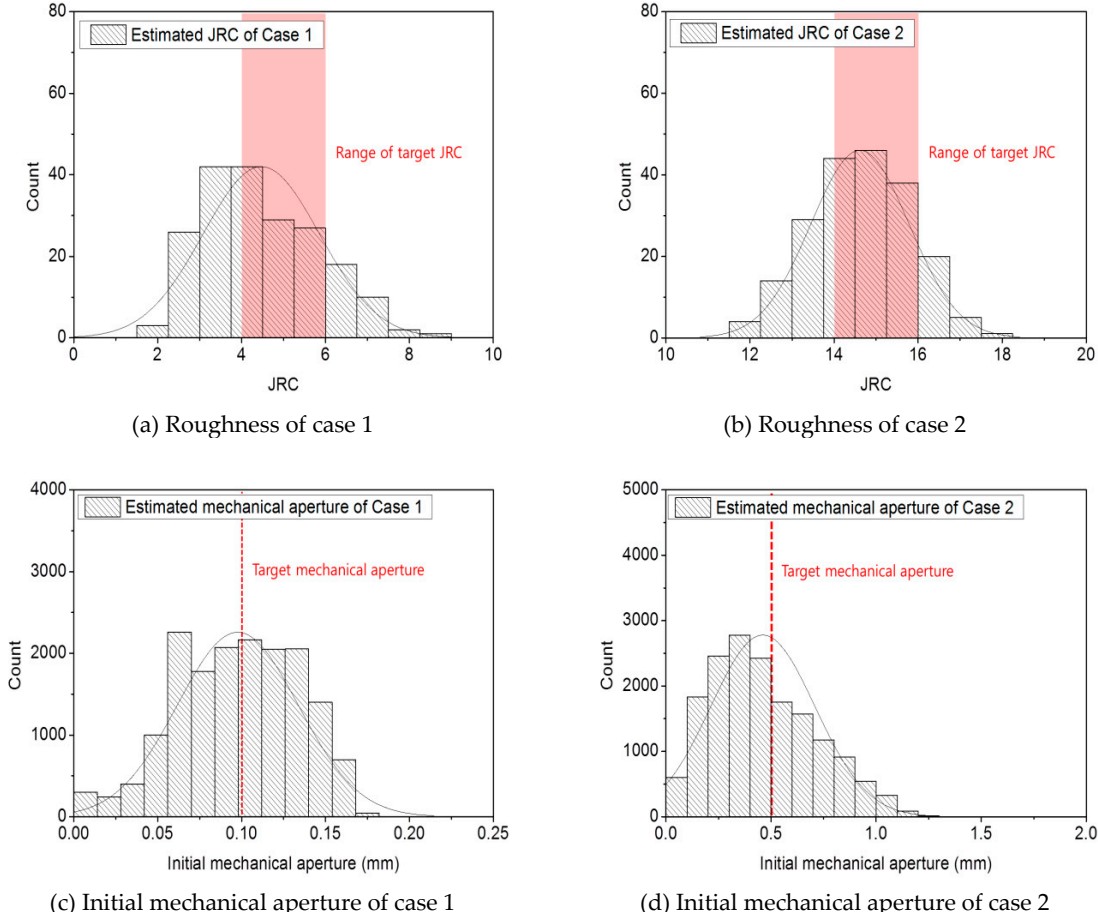

**Figure 8.** Estimated joint roughness (*JRC*) and initial mechanical aperture.

**Table 2.** Estimated results and target values of joint properties.

| | Roughness (*JRC*) | | | Initial Mechanical Aperture (mm) | | |
|---|---|---|---|---|---|---|
| | Target Value | Estimated Results | | Target Value | Estimated Results | |
| | | Ave. | S.D. | | Ave. | S.D. |
| Case 1 | 4~6 | 4.47 | 1.39 | 0.1 | 0.0977 | 0.0360 |
| Case 2 | 14~16 | 14.62 | 1.17 | 0.5 | 0.4611 | 0.2481 |

The method described above dealt with the generation of a single joint. However, analyses on the characteristics of the rock mass, such as DFN (Discreet Fracture Network), may require multiple joints being intersected or branched. Hakami and Stephansson (1993) [29] noted that an intersection of joints could play a significant role in the flow pattern of joint networks, even though the pore volume of intersections, are small compared to total rock mass porosity [29]. Additionally they also provided possible intersection geometries and modelling concepts. In the modelling procedure, the proposed surface-generating method in this study could be applied. Two corresponding surfaces, which are supposed to be in contact, can be generated following the method, and other than that, the rotation of the surface coordinates and the trimming points in conjunction would be required to make the intersections. Though it is out of research scope of this study, it would be promising in future studies.

## 5. Conclusions

A rock joint is an important feature in a rock mass that highly affects its mechanical and hydraulic properties. The joint also has several properties, such as roughness, mechanical aperture, stiffness

and so on. They define the mechanical and hydraulic characteristics of the joint interacting with each other. Roughness and aperture have been drawn attention, since they exert great influence on both the mechanical and hydraulic features of the joint. However, controlling them at the laboratory scale is sometimes consumptive in terms of specimen preparation, since it requires inevitable trial and error approaches.

In this study, a new and simple method for generating an artificial joint specimen was proposed. The specimen included both the upper and lower surfaces and a void space inside. Among several joint generating algorithms, fractal theory, specifically the random midpoint displacement method, was adopted to generate the specimen. By manipulating the input parameters of the algorithm, distributions of roughness and mechanical apertures of the joint could be controlled in a quantitative manner.

A parametric study to estimate the roughness and aperture were conducted. As a result, the roughness increased with an increase in the amplitude of asperity ($\sigma$), and decreased with the increase of the Hurst exponent ($H$). With some predetermined testing environmental factors, such as the length of the joint and the generation level, the roughness could be controlled and manipulated quantitatively. Furthermore, it was found that $H$ had a larger influence on the microscopic asperity, while $\sigma$ had a larger influence on the macroscopic height of asperity. At the same time, the mechanical aperture, which means the separation distance between the corresponding joint surfaces, increased with the increase of $\sigma$, and decreased with the increase of $H$ as well.

In order to validate the proposed method, two joint specimens were generated and printed by a 3D printer with the desired target joint properties. Then, the 3D laser scanning on the printed specimens was conducted to obtain surface information, and further, to calculate the roughness and mechanical aperture. By comparing the target values and estimated values, it was found that the proposed method was capable of generating joint specimens with satisfactory precision. It is anticipated that the proposed method can provide an effective and quantitative procedure for specimen preparation, which can be utilized in theoretical and numerical studies on the joint, and even in experimental works when using recent 3D printing technologies.

**Author Contributions:** Conceptualization, S.C.; investigation, S.C., S.L., and H.J.; supervision S.J.; writing original draft, S.C.; review and editing, S.J.

**Acknowledgments:** This research was supported by the National Strategic Project—Carbon Upcycling of the National Research Foundation of Korea (NRF), funded by the Ministry of Science and ICT (MSITT), the Ministry of Environment (ME), and the Ministry of Trade, Industry and Energy (MOTIE) (NRF—2017M3D8A2085654), and the Korea Agency for Infrastructure Technology Advancement under the Ministry of Land, Infrastructure and Transport of Korean Government (Project Number: 13 Construction Research T01). The institute of Engineering Research at Seoul National University (SNU) provided research facilities for this work. The authors are grateful for the support.

**Conflicts of Interest:** The authors declare no conflict of interest.

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
