# Peer review of "Development of a New Method for the Quantitative Generation of an Artificial Joint Specimen with Specific Geometric Properties"

_sustainability, doi:10.3390/su11020373_

Round 1

Reviewer 1 Report

The comments can be found in the attachment.

Author Response

1. Move the sentence “If the midpoint locates on a side line of the …” on Page 3 Line 106 in the bottom of the page. Otherwise, the expression of the normal distribution is given suddenly.

Response: Thanks for the comments. We relocated the sentence as you suggested.

2. Is there possible to generate an intersecting or branched joint? If not, what’s the difficulties? If yes, the authors should add some discussions and examples.

Response: Thank you for your valuable suggestion. It seems possible. For instance, a simple two-dimensional case can be considered as follows. Each surface is generated segment by segment. In the generation, the method proposed in our paper should be considered when generating two corresponding surfaces, which is supposed to be in contact. Therefore, total 3 pairs of surfaces will follow the designated geometric conditions separately. Other than that, miscellaneous manipulations, such as rotation and trimming would be required. As a possible application, we added statements in Chapter 4.

3. According to Fig. 7, the specimen is separated by a rough joint. The reviewer would like to see an example of an intact specimen embedded a joint rather than two separate parts.

Response: Thanks for the comment. We added pictures of intact and joint-embedding specimen.

4. Here is a suggestion. In oil industry, there are two types of faults. One works a fluid flow conduit with high permeability. The other one is sealing fault working as a barrier for fluid flow. The authors may think about how to fill the void with some other materials to decrease the permeability, which would be very promising.

Response: Thanks for your suggestion. The simplest way is just adding another layer between the upper and lower surfaces, which can represent the infilling material. Depending on the infilling material concerned with, a granular (i.e. fine sand) or powder (i.e. clay or bentonite) layer would be placed to. We are considering your comment in our future research.

Reviewer 2 Report

Interesting paper, I enjoyed reading it.

1) The comparison in Figure 8 and Table 2 is quite confusing. You mentioned that JRC is not really a quantitative parameter, but still, what are you actually comparing then? Can you transform your results into JRC? Or can you transform JRC into another parameter which you can then use for your actual comparison? The descriptions should be something like “target” and “result” with two different colors or something like that.

2) You have talked about the problems with JRC yourself. Why not laserscan a real joint and compare it to the laser scan of your generated joint? I will not insist that you carry out new experiments, but my point is: I would like to get a better impression whether your modeling is actually realistic (compared to  rock joints) or just a random spiky surface. This is an improvement to the paper I would like to see.

Line 28 – avoid „huge“

Figures such are Figure 4 should have larger font size and be provided as vector graphics

Author Response

1) The comparison in Figure 8 and Table 2 is quite confusing. You mentioned that JRC is not really a quantitative parameter, but still, what are you actually comparing then? Can you transform your results into JRC? Or can you transform JRC into another parameter which you can then use for your actual comparison? The descriptions should be something like “target” and “result” with two different colors or something like that.

Response: Thank you for the comment. Obviously any transformation between roughness parameters is possible since we already obtained geometric information of surfaces. The reason we chose to use JRC as a roughness estimator is its broad applicability. As you noted, JRC is not a quantitative parameter but still it is even more frequently used than other parameters in rock mechanics. Figure 8 and Table 2 do not present comparison in fact. They showed estimated JRC and mechanical aperture of printed specimens. Considering the comment, we revised Figure 8 and Table 2 so as to make comparisons between ‘target’ and ‘results’ clearer.

2) You have talked about the problems with JRC yourself. Why not laserscan a real joint and compare it to the laser scan of your generated joint? I will not insist that you carry out new experiments, but my point is: I would like to get a better impression whether your modeling is actually realistic (compared to rock joints) or just a random spiky surface. This is an improvement to the paper I would like to see.

Response: Thanks for your comment. As we noted in the manuscript, making real rock joints is difficult and consumptive. And what we wanted to do is introducing a surface generating method without real joint. So we understood your comment as whether the synthetic joint surface is suitable to represent real rock joint, not comparing the real and generated joints. Roughness of a rock joint can be successfully simulated by fractal theory and many papers have reported the applications. We added more descriptions about it in Chapter 2 to make better impression. Also, the distribution of generated roughness and mechanical aperture shows similar characteristics with the real rock joint, which can support the reality and applicability of our modelling. In this regard, some emphases were added in Chapter 4.

Line 28 – avoid „huge“

Response: We removed the word as you suggested

Figures such are Figure 4 should have larger font size and be provided as vector graphics

Response: We revised Figure 4 to have larger font size and resolution in the manuscript.
